# Establishing Raw Acceleration Thresholds to Classify Sedentary and Stationary Behaviour in Children

**DOI:** 10.3390/children5120172

**Published:** 2018-12-19

**Authors:** Liezel Hurter, Stuart J. Fairclough, Zoe R. Knowles, Lorna A. Porcellato, Anna M. Cooper-Ryan, Lynne M. Boddy

**Affiliations:** 1Physical Activity Exchange, Department of Sport and Exercise Sciences, Liverpool John Moores University, Liverpool L3 2EX, UK; Z.r.knowles@ljmu.ac.uk (Z.R.K.); L.m.boddy@ljmu.ac.uk (L.M.B.); 2Department of Sport and Physical Activity, Edge Hill University, Ormskirk L39 4QP, UK; Stuart.fairclough@edgehill.ac.uk; 3Public Health Institute, Faculty of Education, Health and Community, Liverpool John Moores University, Liverpool L2 2QP, UK; L.a.porcellato@ljmu.ac.uk; 4School of Health and Society, Salford University, Manchester M6 6PU, UK; A.m.cooper-ryan@salford.ac.uk

**Keywords:** accelerometers, wearable technology, activity classification, cut points

## Abstract

This study aimed to: (1) compare acceleration output between ActiGraph (AG) hip and wrist monitors and GENEActiv (GA) wrist monitors; (2) identify raw acceleration sedentary and stationary thresholds for the two brands and placements; and (3) validate the thresholds during a free-living period. Twenty-seven from 9- to 10-year-old children wore AG accelerometers on the right hip, dominant- and non-dominant wrists, GA accelerometers on both wrists, and an activPAL on the thigh, while completing seven sedentary and light-intensity physical activities, followed by 10 minutes of school recess. In a subsequent study, 21 children wore AG and GA wrist monitors and activPAL for two days of free-living. The main effects of activity and brand and a significant activity × brand × placement interaction were observed (all *p* < 0.0001). Output from the AG hip was lower than the AG wrist monitors (both *p* < 0.0001). Receiver operating characteristic (ROC) curves established AG sedentary thresholds of 32.6 mg for the hip, 55.6 mg and 48.1 mg for dominant and non-dominant wrists respectively. GA wrist thresholds were 56.5 mg (dominant) and 51.6 mg (non-dominant). Similar thresholds were observed for stationary behaviours. The AG non-dominant threshold came closest to achieving equivalency with activPAL during free-living.

## 1. Introduction

Sedentary behaviour refers to a complex set of behaviours taking place in a range of settings [1,2] and is defined as any waking, sitting, reclining or lying behaviour with low energy expenditure (≤1.5 Metabolic Equivalent of Tasks (METs)) [3]. Children have a higher resting energy expenditure than adults, therefore 2 METs is the recommended upper boundary for children’s sedentary behaviours [4]. Recently the Sedentary Behaviour Research Network published consensus definitions for terms relating to sedentary behaviour. Standing still is one activity that has received attention, as it cannot be classed as sedentary behaviour, but falls under the category of stationary behaviour. Stationary behaviour is defined as “any waking behaviour done while lying, reclining, sitting, or standing, with no ambulation, irrespective of energy expenditure” [3]. The metabolic cost of standing still is thought to be low [5] and it has been recommended, just as for prolonged seated positions, that standing still for long periods should also be avoided [6], despite this, many studies to date have not examined stationary time.

Recently, sedentary behaviour has received increased attention as an independent health risk factor. Evidence suggests that sedentary behaviour contributes to a number of health-related conditions in children, independent of physical activity [7,8,9]. Accurate monitoring of not only volume, but also types of sedentary behaviour is important before the focus can shift to that of developing effective intervention strategies [10]. Accelerometers are widely accepted objective methods of monitoring children’s physical activity levels [11] and sedentary behaviour [12]. However, most accelerometers are unable to differentiate between postures and have been found to both overestimate [13] and underestimate [14] sedentary time. GENEActiv and ActiGraph wrist accelerometers can predict a person’s most likely posture (sit or stand) through a method known as the Sedentary Sphere [15], but this has not been validated in children. Traditionally, researchers have used accelerometer output reduced to proprietary counts, but counts-based data limits comparisons between studies using different brands [16]. Physical activity intensity cut points derived from raw acceleration output have been developed for the GENEActiv and ActiGraph GT3X+ accelerometers [17], making comparisons between these devices and placements (wrist and hip) possible [18] whilst also increasing researcher control over data processing. Although previous studies have attempted to establish raw sedentary thresholds [19,20,21,22,23,24,25], none of these studies focused solely on children’s sedentary behaviours, rather focusing on physical activity or adult populations. Hildebrand, Van Hees, Hansen and Ekelund [17] used Euclidean norm minus one (ENMO), a data reduction method which results in signal vector magnitude (SVM) values not dependent on sampling frequency or epoch length, allowing for easier comparison between studies. Hildebrand and colleagues have published ActiGraph and GENEActiv ENMO thresholds for physical activity [17] and sedentary time [24] generated from a lab-calibration study. The sedentary thresholds were generated using two sedentary ‘stations’ (lying, watching television and sitting, using a computer) within a wider physical activity calibration protocol. The resultant thresholds were subsequently applied to free-living data but demonstrated low accuracy when compared with activPAL data [26]. One potential reason for the reduced performance during free-living activities was that the stations included within the circuit were not representative of the range of sedentary behaviours that children engage in. 

The present study applied the ENMO method to five sedentary activities representative of typical child behaviours. Data collection took place in the school gymnasium, mimicking a laboratory calibration study setting, but increasing the feasibility and ecological validity of the protocol involved. Furthermore, during a subsequent study, the thresholds were applied to free-living data and compared with data from activPAL as the criterion reference. 

The aims of this study were: (1) to compare the raw accelerometer output of ActiGraph (AG) and GENEActiv (GA) accelerometers across different placements; (2) to identify raw acceleration signal thresholds for different sedentary behaviours in children, from both the hip and wrist, using AG and GA; and (3) to validate the thresholds during free-living activities.

## 2. Materials and Methods

### 2.1. Participants

After gaining ethical approval from the Research Ethical Committee of Liverpool John Moores University (16/SPS/056), all year 5 children (*n* = 60, 9–10 years old), from one primary school in Liverpool, England, were invited to participate in the calibration study. After receiving signed informed parental/carer consent and child assent forms, 27 children (17 girls) took part in this study. During a subsequent study (17/SPS/034), 21 children (13 girls, 9–10 years old) from two primary schools in Liverpool were recruited to participate in the free-living study.

### 2.2. Anthropometrics

Participants’ body mass was measured in light clothing without shoes, to the nearest 0.1 kg using electronic scales (Seca, Birmingham, UK). Stature and sitting height were measured to the nearest 0.1 cm using a stadiometer (Leicester Height measure; Seca, Birmingham, UK). Waist circumference was measured at the midpoint between the bottom rib and the iliac crest, to the nearest 0.1 cm using a non-elastic measuring tape (Seca, Birmingham, UK). Sex-specific regression equations [27] were used to predict children’s age from peak height velocity, which is a proxy measure of biological maturation. Participants self-reported their dominant hand.

### 2.3. Sedentary Behaviour

#### 2.3.1. Calibration Study Protocol

Each participant was fitted with six accelerometers: one ActiGraph GT9X (AG; ActiGraph, Pensacola, FL, USA) and GENEActiv (GA; ActivInsights Ltd., Cambridgeshire, UK) monitor on each wrist (next to each other, in no specific or consistent order), an ActiGraph GT3X on the right hip and an activPAL (PAL Technologies Ltd., Glasgow, UK) monitor on the right anterior thigh. All monitors were worn throughout the testing protocol, which involved seven different stations representative of sedentary behaviour and light physical activity (see Table 1 for detailed description of the stations), with three participants rotating between the stations during each session. Before the standing with phone and sitting with tablet stations, participants were asked whether they were familiar with the games involved. All the participants knew the first game, while two participants were unfamiliar with the second game and they were given time to familiarise themselves with it. The activities were performed for five minutes each, in no particular order except for television viewing, which was always completed first in an effort to prevent the television from distracting participants during the other activities. The stations were designed to simulate children’s typical sedentary activities. Participants’ start and end times for each activity were observed with a Garmin Forerunner 235 wristwatch (synchronized with the same computer time used to initialise all monitors) and recorded. The researchers observed the participants completing the stations (while standing a few meters away). After each session in the school gymnasium, the participants continued to wear the monitors for at least 10 minutes during school recess. Participants were instructed to play as they normally would during recess, while the researchers observed and videotaped them from the side-line. Observational data is not presented here. The testing protocol lasted between 50 and 70 min per data collection session, with two sessions completed per day.

#### 2.3.2. Free-Living Protocol

As part of a separate study, participants were fitted with three monitors: an AG GT9X and GA (both on the non-dominant wrist, with AG distal to GA) as well as an activPAL on the thigh. They were asked to wear the monitors for two days, only removing the wrist-worn monitors for water-based activities. Participants were given log sheets to record when they removed the monitors. After two days the GA and activPALs were collected, while participants continued to wear the AG as part of a larger study.

### 2.4. Accelerometers

The GA, AG GT9X and GT3X are small, lightweight tri-axial accelerometers with a dynamic range of ±8 g. All GA and AG monitors were initialised with a sampling frequency of 100 Hz. While the AG GT9X looks like a watch and shows the time, it is more appealing to wear on the wrist than the slightly bigger, more cumbersome AG GT3X that is more suitable for the hip. The activPAL is a small, single-site lightweight activity monitor that uses proprietary algorithms to classify an individual’s free-living activity into periods spent sitting, standing and walking. It collects data at a sampling frequency of 20 Hz and the default settings were used within this study.

All GA data were downloaded using GENEActiv PC software version 3.1 and saved in raw format as binary files. AG data were downloaded using ActiLife version 6.13.3, saved in raw format as .gt3x files and converted to time-stamped .csv files for data processing. ActivPAL data were downloaded using activPAL3 version 7.2.32, saved as .datx files and converted to .csv event files for processing. 

### 2.5. Data Reduction

Signal processing of GA .bin files and AG .csv files was completed using R package GGIR version 1.5-17 (https://cran.r-project.org/web/packages/GGIR/). The free-living data were processed using GGIR version 1.5-24. GGIR converts the raw tri-axial acceleration values from GA and AG into one omnidirectional measure of body acceleration using the ENMO metric [28], with negative values rounded up to zero. The ENMO metric is sensitive to poor calibration [28]; however, GGIR autocalibrates the raw tri-axial accelerometer signal in order to reduce the calibration error [29]. Autocalibration was carried out for free-living data, but not for the calibration protocol where data were collected over a short period of time. GGIR further reduces the data by calculating the average values per 1 s epoch. The first and last 30 s of data from each activity were excluded to remove any potential transitional movements. The central four minutes were manually extracted and utilised for analysis. Data from all the participants (27) completing the sedentary activities were used to compare accelerometer output across brands and placements. All resulting values are expressed in milli (10^−3^) gravity-based acceleration units (mg), where 1 g = 9.81 m/s^2^.

In order to generate raw acceleration sedentary thresholds using receiver operating characteristic (ROC) curve analysis, data from activPAL were used as the criterion standard. The activPAL “Event” files provide exact time in seconds when posture changes occur, classifying events into sedentary, stand and step. Using an Excel formula, these files were expanded to second-by-second data, classifying each second into sit/lie (0), stand (1) or step (2). The activPAL files contain duplicate seconds, where two postures occurred during the same second. Our Excel formula chose the posture that the participant transitioned into as the classification for that particular second. This happened 28 times (i.e., 28 s) during the 28 h and 20 min of data used for the analysis. All 27 participants’ data from the sedentary stations were used in this part of the analysis, together with 23 of the participants’ recess data. During one data collection session, cold weather prohibited three participants from going outside to play during recess and on another occasion, one participant’s activPAL fell off. The resultant second-by-second activPAL files were synchronised with the 1 s ENMO values from ActiGraph and GENEActiv. ActivPAL data were coded in two different ways: Sit/Lie (0) vs. Stand/Step (1) and Sit/Lie/Stand (0) vs. Step (1). 

During the free-living period, all valid hours between 7:00 and 21:00 on the second day of data collection were included in the analysis. Hours were deemed invalid when the monitors were removed for any number of minutes during that hour, according to the log sheets. Data files were visually inspected using ActiGraph, GENEActiv and activPAL software, to verify recorded log sheet wear time [30]. Thirty-one hours were excluded due to non-wear, while two participants’ activPALs fell off resulting in another 18 h being excluded. 

### 2.6. Data Analysis

Factorial repeated-measures analysis of variance (ANOVA), with Bonferroni corrections were undertaken to determine whether there were differences in output between brands (AG and GA) and placements (dominant- and non-dominant wrists) (interaction effect, brand x placement) for each activity on the circuit. Effect sizes are reported as partial eta-squared (η^2^), with 0.02, 0.13 and 0.26 defined as small, medium and large respectively [31]. Separate one-way repeated measures ANOVAs were undertaken to compare output from the AG hip and wrist monitors. Where assumptions of sphericity were violated, the conservative Greenhouse–Geisser corrected values of the degrees of freedom were used. Only data from the sedentary stations were used for this part of the analysis.

ROC curve analyses were used to identify raw acceleration sedentary and stationary thresholds, from the whole data collection session (sedentary stations and the recess data), with the activPAL data used as the criterion reference standard. To maximise both sensitivity and specificity, the Youden index (*J*; Perkins and Schisterman [32]) was used to identify thresholds. Two ROC curves were generated for each of the five monitors used: the first one was to distinguish between sedentary and non-sedentary behaviours (i.e., sit/lie vs. stand/step), while the second one distinguished between stationary and active behaviour (i.e., sit/lie/stand vs. step). 

In addition, all free-living seconds with a corresponding accelerometer output below the developed thresholds were coded as either sedentary or stationary, while all other seconds were coded as non-sedentary or non-stationary. Agreement between sedentary time according to the thresholds and time spent sitting according to activPAL was examined using paired t-tests and effect sizes calculated as Cohen’s *d* [31] with 0.2, 0.5 and 0.8 defined as small, medium and large. The same was done with the stationary time according to the stationary thresholds and time spent sitting plus standing according to activPAL. Bland–Altman plots compared AG and GA data with that of activPAL. 95% limits of agreement were calculated by mean difference ±1.96 standard deviation of the differences [33]. Free-living data are expressed in minutes or as percentage of total wear time. Furthermore, we also report the following, as recommended by DeShaw et al. [34]: Pearson product correlations, mean percent errors (MPE), mean absolute percent errors (MAPE), and equivalence testing, all as described by DeShaw and colleagues [34].

Statistical analyses were performed using IBM SPSS, version 24 (IBM, Armonk, UK), with the level of statistical significance set at *p* ≤ 0.05 and Microsoft Excel 2016 (Microsoft, Redmond, WA, USA).

## 3. Results

### 3.1. Description of the Population

Descriptive data for all participants are shown in Table 2. Mean anthropometric measurements of the two samples were similar, with only slightly higher waist circumference and body mass observed in the free-living sample.

### 3.2. Comparison of Activities, Accelerometer Brands and Placements 

A factorial repeated measures ANOVA showed a significant main effect of activity on accelerometer output (F_1.47, 9548_ = 18,279; *p* < 0.0001; η^2^ = 0.74). Pairwise comparisons revealed significant mean differences between most activities (all *p* < 0.0001, except standing with phone was higher than homework *p* = 0.001, and TV viewing was significantly higher than standing with phone *p* = 0.003). No significant difference was found between resting and sitting with tablet (*p* = 0.655). Table 3 shows mean accelerometer output from both wrists and both brands, for each activity. A significant main effect of brand was found (F_1, 6479_ = 36; *p* < 0.0001; η^2^ = 0.006), with output from GA slightly higher than AG (mean difference = 1.44, standard error (SE) = 0.24, 95% confidence interval (CI) [0.97–1.91]). A non-significant main effect of placement (dominant and non-dominant wrists) (*p* = 0.259) was observed. However, individual two-factor repeated measures ANOVAs for each activity showed significant main effects of placements (dominant and non-dominant wrists) for all activities except for TV viewing (*p* = 0.321). When analysing ActiGraph data separately (hip and wrists), a significant main effect of placement was found (F_1.97, 12761_ = 2343; *p* < 0.0001; η^2^ = 0.266).

A significant three-way interaction effect (activity × brand × placement) was observed (F_1.77, 11489_ = 16.8; *p* < 0.0001; η^2^ = 0.003). Separate analyses per activity showed significant interactions between brand and placement (dominant and non-dominant wrists) for all the activities except TV viewing (*p* = 0.145) and walking (*p* = 0.293): homework (F_1, 6479_ = 119; *p* < 0.0001), LEGO^®^ (F_1, 6479_ = 122; *p* < 0.0001), resting (F_1, 6479_ = 50.2; *p* < 0.0001), sitting with tablet (F_1, 6479_ = 10.8; *p* < 0.0001), standing with phone (F_1, 6479_ = 17.1; *p* < 0.0001).

Output from the AG hip monitors were significantly lower than the AG dominant (*p* < 0.0001) and non-dominant wrist monitors (*p* < 0.0001). Overall there was no significant difference found between the two wrist placements for both devices (*p* = 0.259), but analysing the activities individually showed significantly higher output from the dominant wrist during homework, LEGO^®^, resting, sitting with tablet and standing with phone (all with *p* < 0.0001) compared to non-dominant wrist, while no significant difference between wrists was observed while TV viewing (*p* = 0.32) and a significantly higher output from non-dominant wrist during Walking (*p* < 0.0001). 

During four activities, the GA wrist monitors produced a significantly higher output than the AG wrist monitors: homework (*p* < 0.0001), walking (*p* < 0.006), LEGO^®^ (*p* < 0.0001), and sitting with tablet (*p* = 0.032). With the exception of homework, these were also the activities with the highest overall mean accelerometer output. The opposite was observed for the other three activities, with AG wrist outputs significantly higher than GA for: resting (*p* < 0.0001), standing with phone (*p* < 0.0001) and TV viewing (*p* = 0.003). Table 4 shows the mean accelerometer output from ActiGraph (AG) and GENEActiv (GA) monitors for all placements across the seven stations, with symbols indicating significant differences between placements and brands from each activity.

### 3.3. Threshold Generation

Table 5 shows the results from the ROC curve analysis, with the developed sedentary and stationary thresholds. Thresholds for the hip monitors are lower than for wrist-worn monitors. Classification accuracy was significantly better than chance for sedentary and stationary ROC curves. Classification accuracy was however lower for sedentary behaviour (area under the curve (AUC) 0.746–0.797), in comparison to stationary behaviour (AUC 0.888–0.944). Sensitivity was high for all the thresholds identified (>80%), but specificity was lower for the sedentary thresholds (51%–60%). Whereas, specificity for the stationary thresholds was higher ranging from 85%–89%. 

Similar acceleration thresholds were identified for sedentary and stationary behaviours, with the exception of the non-dominant wrist placements (both AG and GA) which found slightly higher thresholds for classifying stationary behaviour.

### 3.4. Validation of Thresholds during Free-Living Time

During the free-living period, mean wear time was 700 ± 176.9 min (11.7 h). Participants spent on average 67% (466.3 ± 131.9 min) of their time seated according to activPAL. The corresponding estimates of sedentary time according to the developed sedentary thresholds were both significantly higher (AG: 71%, 499.5 ± 143.1 min, *p* = 0.003, *d =* 0.25 and GA: 73%, 509.8 ± 145 min, *p* < 0.001, *d* = 0.33). Conversely, estimates of stationary time according to the developed stationary thresholds were both significantly lower (AG: 75%, 522.1.1 ± 147.6 min; *p* < 0.001; *d* = 0.46 and GA: 76%, 529.6 ± 148.5 min, *p* < 0.001, *d* = 0.4) compared to time spent sitting/lying plus standing according to activPAL (85%, 594.6 ± 161.2 min). Table 6 summarises the various indicators of measurement agreement between the two brands against the reference, activPAL. On average, AG overestimated sedentary time by 4% compared to activPAL, with a computed MPE of −7.3 and MAPE of 9.5. Similarly, GA overestimated sedentary time by 6%, with a computed MPE of −9.5 and MAPE of 10.6. AG on average underestimated stationary time by 10% compared with activPAL, with the same computed MPE and MAPE values (both 12.2%). GA underestimated stationary time by 9%, with both MPE and MAPE values of 10.9%. Correlations with activPAL were high for sedentary time (both brands: *r* = 0.95, *p* < 0.001) and stationary time estimates (both brands *r* = 0.98, *p* < 0.001). Figure 1 and Figure 2 show Bland–Altman plots assessing the agreement between sedentary time from activPAL and sedentary time from AG (1A) and GA (1B) and agreement between stationary time from activPAL and stationary time from AG (2A) and GA (2B). The sedentary thresholds had smaller mean biases (AG = +33 min; GA = +44 min) than the stationary thresholds (AG = −72 min; GA = −65 min). Both sedentary thresholds had wider limits of agreement (AG: from −54 to +120 min, GA: from −44 to +132) than the stationary thresholds (AG: from −141 to −4 min, GA: from −124 to −6 min). 

Figure 3 shows the equivalence zones for sedentary (top figure) and stationary (bottom figure) time estimates from AG and GA compared to activPAL. While none of the estimates were found to be statistically equivalent on average at the group level to activPAL when using 10% of the activPAL mean as the zone of equivalence, the AG sedentary threshold of 48 mg was closest to achieving group-level equivalence. The figures clearly show that the sedentary thresholds slightly overestimated, while the stationary thresholds underestimated time spent sedentary or stationary in comparison with activPAL. 

A significant difference was found between AG and GA sedentary time (mean difference = −10.4 min, SE mean = 3.4, *p* = 0.006), however this difference yielded a small effect size of *d* = 0.07. Similarly, a significant difference was found between AG and GA stationary time (mean difference = −7 min, SE mean = 2.6, *p* = 0.01, *d* = 0.05). Figure 4 shows Bland–Altman plots assessing the agreement between sedentary and stationary time from AG and GA. Both had small mean biases (sedentary time = +10.4 min; stationary time = +7.5 min), and narrow limits of agreement (sedentary time −19.8 min to +40.5 min; stationary time from −16 min to +30.9 min). 

## 4. Discussion

The first aim of this study was to compare the raw accelerometer output of AG and GA accelerometers across three different placements. The significantly lower output observed from the AG hip monitors (compared with wrists) is consistent with previous findings [18,24,35]. These results suggest that sedentary behaviour studies should not compare data collected using hip monitors to ones using wrist monitors unless effective harmonisation approaches are used. 

Inconsistent differences between monitors and placements were observed across the range of sedentary and stationary activities. The differences by activity could be attributed to the nature of the sedentary behaviours themselves. While the unique nature of children’s physical activity patterns have been well established [36], little is known about differences in their sedentary behaviours. For example, the homework and standing with phone stations overall had the lowest average accelerometer output of all the stations, even lower than television viewing and resting. Even though statistically significant, many of these differences were small, as seen in Table 3 and Table 4. With the exception of walking and LEGO^®^, the mean differences in output between activities ranged from 0.88 mg to 7.48 mg, differences that are unlikely to be meaningful. A possible explanation for the inconsistencies observed between monitors might be internal differences between devices. The difference observed between brands, although statistically significant, yielded a small effect size, which is unlikely to be meaningful. The dominant wrist monitors produced a higher output for most of the activities. Previous studies have used either the non-dominant wrist [18,37] or the dominant wrist [38,39], with few studies comparing outputs between these two placements. Two studies that compared wrist placements [19,20] found no significant differences between these two sites; however, these studies did not differentiate between dominant and non-dominant wrists, but rather compared the right side with the left. A recent study [38] argues for the use of the dominant wrist in an effort to capture more activities requiring the use of the dominant hand. 

The second aim of this study was to develop raw acceleration thresholds for sedentary and stationary behaviours in children. The lower classification accuracy (AUC) for differentiating between sedentary and non-sedentary behaviour may be partially attributed to the misclassification of some activities by activPAL. Comparing the direct observation notes with the activPAL data showed that activPAL misclassified sitting as standing on a number of occasions for different participants. For example: for two participants activPAL misclassified the entire duration of the seated homework station (5 min) as standing. Similar inconsistencies were found throughout the data which amounted to a total of 30 min of data. This may have had an impact on the accuracy of the thresholds generated using activPAL and the free-living validation. A possible explanation for this may lie in the nature of children’s sedentary behaviours (not sitting completely still for example). Participants might have been sitting on the edges of their seats, with their legs hanging down and thighs outside of the threshold angle to be classified as sitting by activPAL. Children were instructed to sit, however, researchers did not ask participants to sit with their legs at a specific angle to reflect the ‘typical’ behaviour of each child. The misclassifications observed suggest that the activPAL may underestimate sitting and overestimate standing, which is consistent with findings from another study [40]. Other observations made by the researchers include the participants’ inability to lie still during the resting station. Participants were instructed to lie down and rest, as still as possible. This in practice seemed very difficult for the participants, which explains why the resting station did not result in the lowest accelerometer output, as might have been expected. Rather, it ranks fourth in total accelerometer output, with TV viewing, standing with phone and the homework station resulting in lower total accelerometer outputs. 

The stationary activity ‘standing with phone’, produced the second lowest output from wrist monitors and the lowest output from hip monitors, highlighting the fact that children stand exceptionally still while playing with a mobile phone. Adding “stand” to the second ROC curve analysis (stationary behaviour) resulted in higher classification accuracy (the lowest being the GA dominant wrist with an AUC = 0.888 and highest the AG hip with AUC = 0.944) than when standing was excluded. Using stationary activity also removed the issues associated with the misclassification of activities by the activPAL. Despite this, the resultant thresholds for stationary behaviour are similar to those for sedentary behaviour, except for a slight difference between non-dominant wrists. Using the dominant wrist thresholds in future studies would thus capture both stationary and sedentary behaviours. 

In comparison to those of Hildebrand, Hansen, van Hees and Ekelund [24] our threshold for AG hip monitors is lower (32.6 mg versus 63.3 mg), while our AG non-dominant wrist threshold is higher (48.1 mg versus 35.6 mg). Our GA non-dominant wrist threshold is slightly lower than that of Hildebrand et al. (51.6 mg versus 56.3 mg) who did not include the dominant wrist in their protocol. Sensitivity and specificity were higher in the Hildebrand et al. [24] results, possibly because of protocol differences: Hildebrand used two sedentary activities in a controlled laboratory environment and four light-to-vigorous physical activities. Hildebrand also concluded that posture misclassification from activPAL might have attributed to the lower specificity observed. Another noticeable difference between our results and those from Hildebrand is that our specificity increased greatly when we included the standing activity (e.g., from 57% to 85% for the GA non-dominant wrist monitors), while Hildebrand’s increased when they excluded the standing activity from their analysis. This observation is likely due to protocol differences. For example, during a standing activity in the Hildebrand protocol children were allowed to draw on a whiteboard, which probably resulted in more movement than our standing with a phone station. 

Our GA non-dominant wrist threshold (51.6 mg) is similar to the recently published 51 mg by Boddy, Noonan, Kim, Rowlands, Welk, Knowles and Fairclough [26]. Our sensitivity was slightly higher than that of Boddy et al. [26] (87% vs. 81%), with both studies’ specificity at 57%. There is a 3.5 mg and 3.2 mg difference in our resultant sedentary and stationary thresholds estimated with AG and GA for non-dominant wrist, respectively. As previously stated, these small differences might be the result of internal differences between devices, and unlikely to be meaningful in practice. Future researchers might decide to use 50 mg as sedentary threshold and 60 mg as stationary threshold for both brands, facilitating comparisons between studies. 

For two participants, there were periods classified as sitting by activPAL during recess. Observational data showed that the participants were not sitting, but rather hunching down during play (for example crouching behind an object during hide and seek). While the posture classification from activPAL was correct, the behaviour was not sedentary, which again highlights the differing nature of children’s sedentary behaviours in comparison to adults and the value of observational data. Similar observations were made during an activPAL validation in preschool children [40] whereby the postures lie, sit, stand and walk were too limited for the range of positions children assume during play time. Cumulatively these findings question the suitability of using activPAL to classify what appears to be a wide range of sedentary behaviours performed by children. 

During the free-living period, results from the paired t-tests as well as the various indicators of measurement agreement suggested that the sedentary thresholds performed better than the stationary thresholds compared with data from activPAL. The sedentary cut points for both AG and GA slightly overestimated time spent sedentary compared with time spent sitting/lying according to activPAL. While these were significant differences, the effect sizes were relatively small and equivalence testing showed that the 48 mg AG cut point came close to achieving equivalence with activPAL on average at the group level. Conversely, stationary cut points for both AG and GA underestimated time spent sitting/lying plus standing according to activPAL. For both brands, the computed MPE and MAPE values for the stationary thresholds were the same, confirming that all the error was in one direction (i.e., an underestimation of stationary time). 

The main reason for the differences observed between cut points and activPAL data is that we are essentially comparing a lack of movement (or very little movement) with posture classifications. When using cut points to analyse sedentary behaviour data, researchers should acknowledge that there are certain circumstances that can result in misclassification. For example, where a lack of movement at the wrist will be classified as sedentary using cut points, the participant might in fact be standing. Conversely, the stationary thresholds’ underestimation of stationary time is likely due to children moving their arms while standing. This behaviour is called “active standing” (defined as waking activity characterised by energy expenditure above 2.0 METs, while standing without ambulation [3]). When using cut points to analyse the data, it is unlikely that wrist-worn accelerometers would be able to differentiate between active standing and light-intensity physical activity with ambulation, for example slow walking, meaning this behaviour is incorrectly classified. Achieving an accurate estimate of stationary behaviours appears to be challenging using wrist-mounted accelerometers in the absence of postural information. While we agree with the recently published consensus definitions of sedentary and stationary behaviours [3], we do argue that it is better to misclassify passive standing (like playing with a mobile phone) as sedentary behaviour rather than as physical activity when identifying children that would benefit from intervention; therefore, we recommend the use of the sedentary thresholds in practice. From a health perspective, it is better to overestimate sedentary time than to underestimate sedentary or stationary behaviours, as active children can still participate in interventions aiming to decrease sedentary or stationary time without causing any harm. Conversely, underestimating sedentary time might result in children not being identified for intervention and ultimately exposed to increased health risk. Where more accurate measures of sedentary and stationary time from wrist-worn accelerometers are needed, using postural approaches such as the Sedentary Sphere [15] method in children might be a better option, although validation studies are required to examine this further. 

Our study has several strengths. The protocol included seven different activities, representing a wide range of ‘typical behaviours’ in children, as well as playtime data. It took place in a school gymnasium and outside on the playground, spaces that the participants were familiar and comfortable with. This increased the ecological validity of the protocol involved. The participants wore six different monitors each, and raw data processing as opposed to proprietary counts allowed for various direct comparisons between brands and placements. There are also some limitations: we used a convenience sample and all the participants came from the same school, which might not be representative of the wider population. The homogeneous sample of 9- to 10-year-old children should not be considered representative of all children, and further studies are needed for younger and older populations to expand these findings. The activities were not performed in the same order for each participant, however, no formal randomisation techniques were used. AG and GA monitors were placed next to each other on the wrist, in no specific or consistent order [41]. Placing one brand consistently distal to the other might have resulted in increased acceleration from that brand, however no formal randomisation techniques were used. Whilst activities were designed to reflect children’s typical sedentary behaviours, METs could not be measured and as a result we cannot assume that energy expenditures during the protocol were at all times ≤1.5 METs. The protocol in the school gymnasium highlighted the fact that activPAL sometimes misclassifies children’s postures, and we have to assume that the same might have happened during the free-living period. However, except for direct observation, which was unfeasible for this study, there is no other tool that can be used as a criterion measure. 

## 5. Conclusions

This study has identified raw acceleration sedentary and stationary thresholds for the AG hip, dominant and non-dominant wrists as well as the GA dominant and non-dominant wrists for children. The stationary thresholds underestimated stationary time when applied to free-living data in relation to activPAL. The sedentary thresholds were not comparable; however, effect sizes were small and the AG cut point came close to achieving equivalence with activPAL on average at the group level. Comparisons between accelerometer brands and placements in the calibration study produced inconsistent results; however, the free-living data confirmed that these differences are small. Future studies focusing on the nature of children’s sedentary behaviours may provide insight into the reasons for the differences observed.

## Figures and Tables

**Figure 1 children-05-00172-f001:**
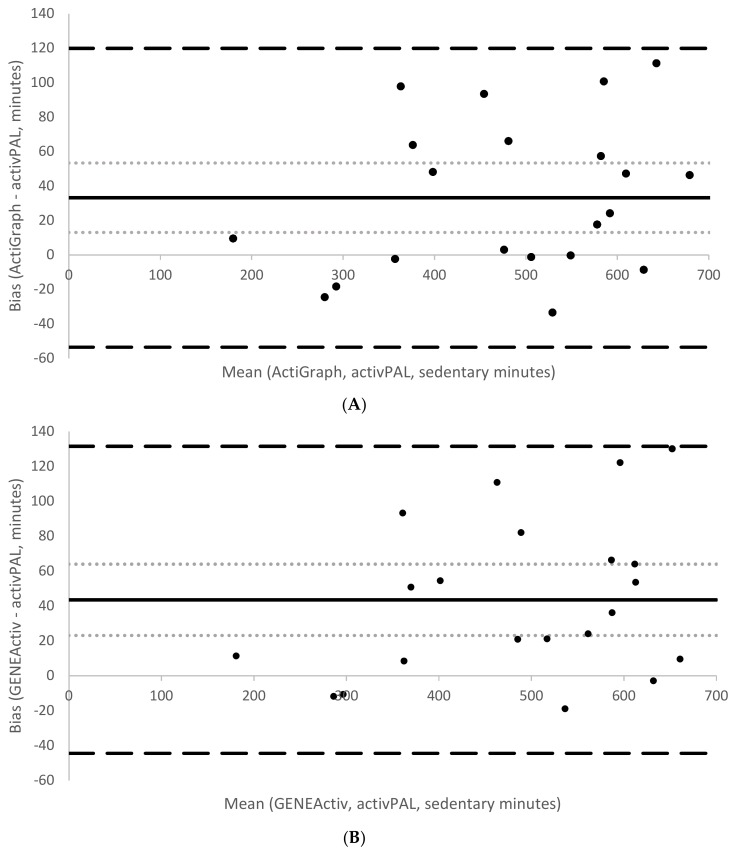
Mean bias (solid line), 95% CI of the mean bias (light dotted lines) and 95% limits of agreement (large dashed lines) for the sedentary free-living time estimated by the developed thresholds for Actigraph non-dominant wrist (**A**) and GENEActiv non-dominant wrist (**B**) relative to activPAL.

**Figure 2 children-05-00172-f002:**
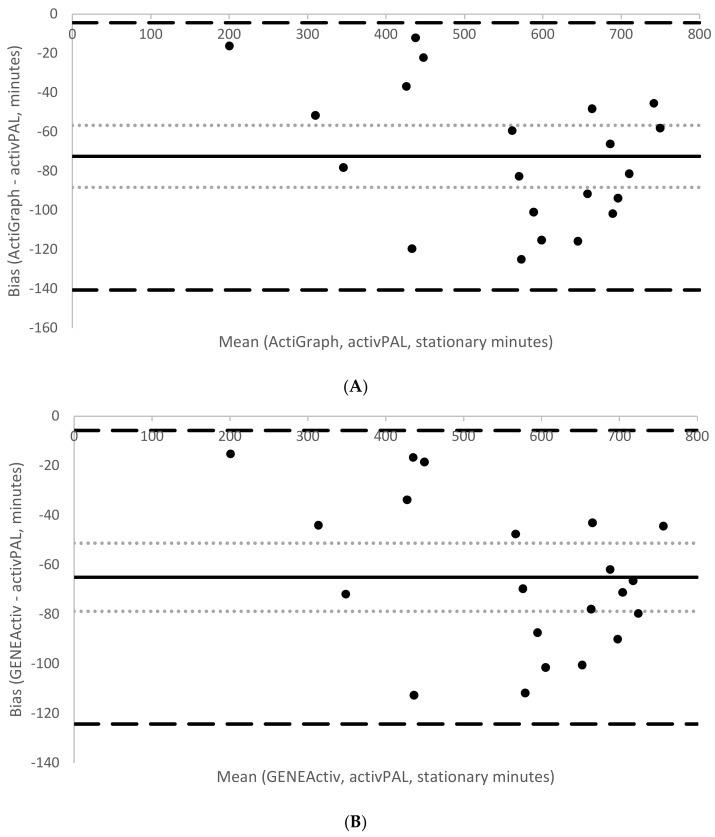
Mean bias (solid line), 95% CI of the mean bias (light dotted lines) and 95% limits of agreement (large dashed lines) for the stationary free-living time estimated by the developed thresholds for ActiGraph non-dominant wrist (**A**) and GENEActiv non-dominant wrist (**B**) relative to activPAL.

**Figure 3 children-05-00172-f003:**
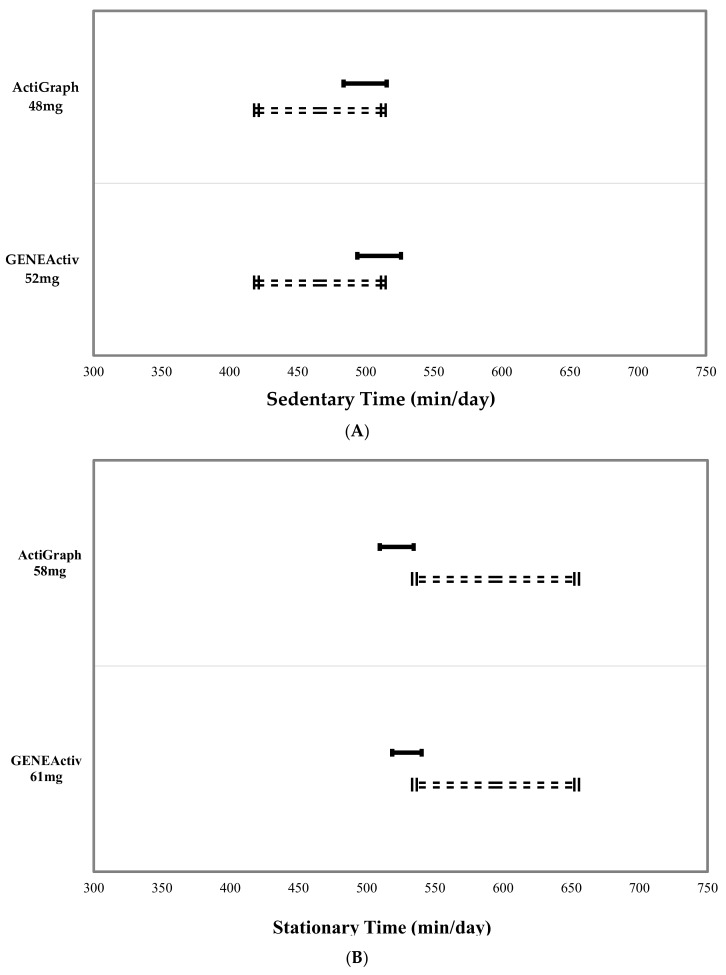
activPAL sedentary (**A**) and stationary (**B**) time zones of equivalence (dotted lines) and 90% confidence intervals for the ActiGraph and GENEActiv sedentary (**A**) and stationary (**B**) time estimates classified using the developed cut points.

**Figure 4 children-05-00172-f004:**
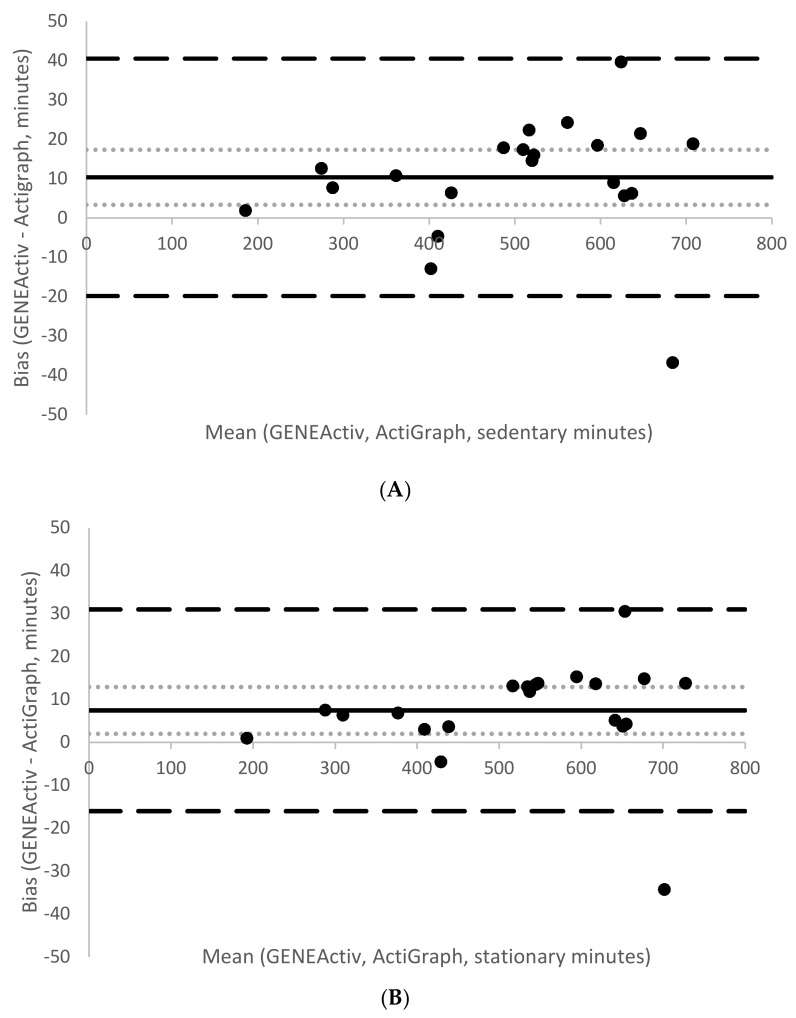
Bland–Altman plots comparing sedentary (**A**) and stationary (**B**) time estimates from the developed cut points between GENEActiv and ActiGraph. The figure displays mean bias (solid line), 95% CI of the mean bias (light dotted lines), and 95% limits of agreement (large dashed lines).

**Table 1 children-05-00172-t001:** Description of the seven sedentary behaviour and light physical activity stations.

Station	Description
Resting	Lying on a soft gym mat, in a supine position, asked to avoid bodily movements.
TV viewing	Sitting comfortably on a couch, watching television.
Seated, tablet	Sitting comfortably on a couch, playing the Bike Race game on an iPad.
Seated, LEGO^®^	Sitting at a table, playing with LEGO^®^.
Seated, Homework	Sitting at a table, copying a piece of writing (mimicking homework).
Standing, phone	Standing while playing Subway Surf on a mobile phone.
Walking	Walking, at own pace, around a designated track.

**Table 2 children-05-00172-t002:** Descriptive characteristics of the participants [mean (standard deviation, SD)].

Variable	Calibration Study (*n* = 27)	Free-Living Data (*n* = 21)
Age (years)	10.2 (0.3)	10.2 (0.3)
Stature (cm)	141.5 (6.9)	142.8 (7.4)
Sitting height (cm)	70.9 (3.9)	71.3 (3.3)
Waist circumference (cm)	66.7 (10.9)	70.3 (9.8)
Body mass (kg)	37.3 (11.4)	40.8 (10.6)
BMI (kg/m^2^)	18.3 (3.9)	19.8 (4)
Maturity offset (years)	-2.3 (1.1)	−2.1 (1)
APHV * (years)	12.3 (1.1)	12.3 (0.6)

* APHV = age from peak height velocity.

**Table 3 children-05-00172-t003:** Mean accelerometer output from both brands and wrists, for all activities, from highest to lowest.

	Mean		95% Confidence Interval
Activity	Acceleration (mg)	Lower Bound	Upper Bound
Walking	190.7	188.4	193.0
LEGO^®^	31.0	30.6	31.4
Seated, tablet	20.5	20.0	21.0
Resting	19.6	19.0	20.2
TV viewing	15.0	14.5	15.5
Standing, phone	13.9	13.5	14.1
Homework	13.0	12.7	13.3

**Table 4 children-05-00172-t004:** Accelerometer output [mean (SD)] from ActiGraph (AG) and GENEActiv (GA) monitors, expressed in mg, across all stations (*n* = 27).

Device	Resting	TV Viewing	Seated, Tablet	Standing, Phone	Seated, LEGO^®^	Seated, Homework	Walking
AG hip	8.9 (12.4) *	5.5 (7.7) *	8.4 (8.6) *	3.9 (8.1) *	6.5 (8.0) *	5.3 (8.5) *	148.2 (51.5) *
AG Dom	23.5 (34.1) †,#	15.2 (27.7)	21.5 (28.7) †	16.7 (19.2) †,#	32.8 (25.7) †,#	13.5 (19.8) †,#	178.0 (139.3) †,#
GA Dom	18.0 (38.6)	14.5 (33.2)	21.4 (32.6) †	14.8 (18.9) †	36.6 (29.9) †	19.0 (23.4) †	183.1 (115.4) †
AG Ndom	18.7 (36.1)	15.8 (27.8) #	18.6 (29.3) #	12.0 (23.5)	21.8 (27.6) #	9.3 (19.1) #	199.3 (131.0)
GA Ndom	17.9 (36.4)	14.4 (27.8)	20.4 (30.8)	11.9 (21.8)	32.6 (31.6)	10.2 (20.5)	202.1 (129.3)

* significantly different from wrists (p < 0.0001), † = significantly different from non-dominant wrists (p < 0.05), # = AG significantly different from GA (p < 0.05).

**Table 5 children-05-00172-t005:** Sensitivity, specificity, area under the curve (AUC) and 95% confidence intervals (CI), with proposed thresholds for ActiGraph (AG) hip, dominant- (Dom) and non-dominant (Ndom) wrists as well as GENEActiv (GA) dominant and non-dominant wrists in children.

Sedentary Behaviour						Stationary Behaviour				
Device	Sensitivity (TPR *)	Specificity (TNR †)	AUC	95% CI	Threshold (mg)	Sensitivity (TPR)	Specificity (TNR)	AUC	95% CI	Threshold (mg)
AG hip	97%	51%	0.746	0.743–0.75	32.6	94%	86%	0.944	0.942–0.946	32.6
AG Dom	89%	55%	0.759	0.756–0.762	55.6	86%	87%	0.926	0.924–0.928	55.2
AG Ndom	87%	60%	0.797	0.788–0.793	48.1	87%	89%	0.940	0.939–0.942	57.5
GA Dom	84%	57%	0.752	0.749–0.755	56.5	82%	85%	0.888	0.886–0.891	59.1
GA Ndom	87%	57%	0.77	0.768–0.773	51.6	86%	85%	0.918	0.916–0.920	60.7

* True Positive Rate † True Negative Rate.

**Table 6 children-05-00172-t006:** Sedentary and stationary time estimates from AG and GA free-living data compared with activPAL.

Criterion	Comparison	Mean (SD) Minutes	MPE (SD)	MAPE (SD)	Limits of Agreement (Lower to Upper)	95% CI of Mean Biases (Lower to Upper)	Correlation	*p* Value	Equivalency Analysis (Minutes)
**Sedentary time**									
activPAL (sit/lie)		466.3 (131.9)							Zone of Equivalence: 419.6–512.9
	ActiGraph (48 mg)	499.5 (143.1)	−7.3% (10.5%)	9.5% (8.5%)	from −54 to 120	from 13 to 53	0.95	0.003	90% CI 483.6–515.3
	GENEActiv (52 mg)	509.8 (145.0)	−9.5% (10.1%)	10.6% (8.8%)	from −44 to 132	from 23 to 64	0.95	<0.001	90% CI 493.8–525.9
**Stationary time**									
activPAL (sit/lie/stand)		594.6 (161.2)							Zone of equivalence: 535.1–654
	ActiGraph (58 mg)	522.1 (147.6)	12.2% (5.6%)	12.2% (5.6%)	from −141 to −4	from −88 to −57	0.98	<0.001	90% CI 509.6–534.5
	GENEActiv (61 mg)	529.6 (148.5)	10.9% (4.9%)	10.9% (4.9%)	from −124 to −6	from −79 to −51	0.98	<0.001	90% CI 518.7–540.4

MPE = mean percentage error; MAPE = mean absolute percentage error; CI = confidence interval.

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
