# Peer review of "Establishing Raw Acceleration Thresholds to Classify Sedentary and Stationary Behaviour in Children"

_children, 2018, doi:10.3390/children5120172_

Round 1

Reviewer 1 Report

The purpose of the submitted manuscript is to investigate the raw accelerometer output generated with ActiGraph and GENEActive devices across three wear locations (non-dominant wrist, dominant wrist and hip) in children and to identify a threshold for the assessment of sedentary and stationary behavior.

The first objective of the present study is the comparison between ActiGraph and GENEActive devices. In the study by Hildebrand et al. (2014) the same objective was investigated and results and conclusion are confirmed in the present study. The difference between hip and wrist placement with raw acceleration as previously demonstrated is also demonstrated with the present study.

The second aim of the submitted study looks like a replica of the study conducted by Hildebrand et al. (2016). The major difference between these studies is the inclusion of additional sedentary activities in the assessment of the threshold and including the dominant wrist with the present study. An adult group was also included in the study by Hildebrand.

It is unclear how the data in table 3 (all devices types and placements??) has been generated. However, the difference in output between the different sedentary activities is obviously based on these estimates. A further look at the output generated and described in table 4 seems to suggests that only the output generated with the LEGO activity that is marginally different from TV Viewing, Standing Phone, Tablet and homework. This suggests that three of the activities included do not add additional information to the estimation of sedentary/stationary thresholds. The activities could potentially be excluded.

The assumption of the manuscript is that stationary activities are associated with a higher accelerometer output as sedentary activities. However, the output generated during the standing phone activity is similar to the sedentary activities, which seems to explain the low specificity for the sedentary ROC analysis and the increased specificity with the stationary ROC analysis. Thus, the increased stationary threshold generated in the study is most likely caused by the LEGO activity and not the inclusion of the standing phone activity. This is rather paradoxical and should be addressed. 

The mean bias (sedentary time during free-living as compared to ActivPAL) demonstrated with the present study as compared to Hildebrand is substantially decreased with all devices types and placements. The free-living validation was 24 hours in the Hildebrand study and two days measurements with the present study and the data reduction used with the free-living data is quite different between the two studies. In the present study only data between 07:00 and 21:00 was included and in the Hildebrand study nothing was excluded. This difference in data reduction could explain the substantial difference between the mean absolute biased observed in the two studies. It seems of great importance that this difference is discussed and the potential interpretation.

Comments to figure 1.

There is a 3.5 and 3.2 mg difference in the sedentary and stationary thresholds estimated with AG and GA for non-dominant wrist respectively. Looking at figure 1 it seems like C and D are almost identical whereas the scatter for A and B is quite differently. The small difference between the thresholds in combination with the approximately 10 mg increase from sedentary to stationary seems to require some explanation. It seems rather strange that 10 mg increase with the sedentary to stationary thresholds change the mean bias between brands from highly correlated to highly un-correlated. This strange phenomenon should be discussed.

Author Response

Reviewer 1 comments

Response

It is unclear how the   data in table 3 (all devices types and placements??) has been generated.   However, the difference in output between the different sedentary activities   is obviously based on these estimates. A further look at the output generated   and described in table 4 seems to suggests that only the output generated   with the LEGO activity that is marginally different from TV Viewing, Standing   Phone, Tablet and homework. This suggests that three of the activities   included do not add additional information to the estimation of   sedentary/stationary thresholds. The activities could potentially be   excluded.

These are the estimated marginal means from both brands   and both wrists (not the hip data), for each activity.

We added the following to clarify: “from both wrists and   both brands” to Line 213 and Line 247 (Table heading), as well as   “acceleration (mg)” in the table   itself.

A strength of the study is that the different activities   reflect the range of activities children engage in, therefore, we have   included all data. It is an interesting finding that the output from   accelerometers for the activities are similar and of relevance to the   discipline

The assumption of the   manuscript is that stationary activities are associated with a higher   accelerometer output as sedentary activities. However, the output generated   during the standing phone activity is similar to the sedentary activities,   which seems to explain the low specificity for the sedentary ROC analysis and   the increased specificity with the stationary ROC analysis. Thus, the   increased stationary threshold generated in the study is most likely caused   by the LEGO activity and not the inclusion of the standing phone activity.   This is rather paradoxical and should be addressed.

This is a valid point, however,   we did not make any assumptions with regards to the thresholds prior to   analysis. The relevance of stationary behaviour is related to energy   expenditure and the definition of sedentary behaviour. A common criticism of   accelerometers whether placed at the wrist or hip is that stationary   behaviour is misclassified as sedentary behaviour. Our protocol allowed us to   examine this, and come up with thresholds to classify this behaviour- which   would not necessarily be higher than those for sedentary behaviours.

The data generated from 10 minutes of recess was also   included in the calculation of the thresholds, which added more standing   (stationary) and active time and was reflective of typical behaviours / patterns   of behaviour.

The mean bias   (sedentary time during free-living as compared to ActivPAL) demonstrated with   the present study as compared to Hildebrand is substantially decreased with   all devices types and placements. The free-living validation was 24 hours in   the Hildebrand study and two days measurements with the present study and the   data reduction used with the free-living data is quite different between the   two studies. In the present study only data between 07:00 and 21:00 was   included and in the Hildebrand study nothing was excluded. This difference in   data reduction could explain the substantial difference between the mean   absolute biased observed in the two studies. It seems of great importance   that this difference is discussed and the potential interpretation.

This is a valid point, but we have a different   interpretation of the Hildebrand protocol. Hildebrand states that she   excluded sleep from her analysis of the free-living data (though she doesn’t   say exactly how she excluded it), and had a mean wear time of 791 min (SD   172) which is ~91 mins higher than our mean wear time of 700 min (SD 176.9).

Comments to figure 1.

There is a 3.5 and 3.2   mg difference in the sedentary and stationary thresholds estimated with AG   and GA for non-dominant wrist respectively. Looking at figure 1 it seems like   C and D are almost identical whereas the scatter for A and B is quite   differently. The small difference between the thresholds in combination with   the approximately 10 mg increase from sedentary to stationary seems to   require some explanation. It seems rather strange that 10 mg increase with   the sedentary to stationary thresholds change the mean bias between brands   from highly correlated to highly un-correlated. This strange phenomenon   should be discussed.

We have now added additional statistical approaches to   compare the brands with activPAL and each other, as suggested by Reviewer 2.

Figure 1: A &B. The scale has been edited so the same   scale and axis labels have been used. Figures A & B now appear   consistent, with a similar pattern of points displayed.

We have also added the following sentences in the   discussion:

Line 396-401: “There is a 3.5mg and 3.2mg difference in our resultant sedentary and   stationary thresholds estimated with AG and GA for non-dominant wrist   respectively. As previously stated, these small differences might be the   result of internal differences between devices, and unlikely to be meaningful   in practice. Future researchers might decide to use 50mg as sedentary threshold and 60mg as stationary threshold for both brands, facilitating   comparisons between studies.”

Apologies, we are not clear   what the reviewer is referring to when mentioning bias changes from highly   correlated to highly un-correlated.  The   newly added equivalence testing shows that the sedentary time estimates are   not far from being equivalent to activPAL, and while the results from the   t-tests did show a significant difference between sedentary time estimates   and activPAL, the equivalence testing is more appropriate as it is powered to   test agreement instead of differences (DeShaw, K.J.; Ellingson, L.; Bai, Y.;   Lansing, J.; Perez, M.; Welk, G. Methods for activity monitor validation   studies: An example with the fitbit charge. Journal for the Measurement of   Physical Behaviour 2018, 1, 130-135).

Reviewer 2 Report

SUMMARY

The present study, conducted on children only, had several aims. A first aim was to compare raw acceleration outputs from accelerometers that were (i) of different brands (GENEActiv and ActiGraph), (ii) worn at different body locations (dominant/non-dominant wrists and hip), and (iii) worn during various sedentary/stationary activities and physical activities. A second aim was to develop raw acceleration signal cut-points to classify sedentary, stationary, and active behaviours. Finally, a third aim was to validate these cut-points in free-living setting.

The main contribution of this study is to propose raw acceleration cut-points (and respective accuracy) both to discriminate sedentary behaviours from non-sedentary behaviours and to discriminate stationary behaviours from active behaviours in children, in particular using activities commonly encountered in children. Indeed, most of previous studies conducted in children developed cut-points based on METs data only, not on activity/posture types.

GENERAL COMMENTS

The main strength of this study is that it seems to be the first to develop raw acceleration thresholds to discriminate both sedentary/non-sedentary behaviours and stationary/active behaviours in children, with a validation study conducted in free-living setting. In particular, very few studies have investigated stationary behaviour, that is a concept that has been recently formalized in the literature. Moreover, this study includes comparisons of several accelerometers and body locations, which could allow future users and researchers to choose the most suited/accurate option depending on their objectives and constraints.

This study also presents important limits. One of the major limits is that the reference used to develop the cut-points was the activPAL device, that did not consistently and correctly classified actual behaviour. As the authors were present to observe participants during the calibration study, it should be explained why direct observation was not used as reference to develop the cut-points. Indeed, while the method used in the study allows comparisons with previous studies that also used the activPAL, the accuracy and thus the interest of these cut-points appears limited.

Moreover, only walking was used as structured active behaviour during the calibration study, while it could be expected that other active behaviours are used to correspond to children physical activity patterns.

Finally, it should be clarify whether the sedentary behaviours proposed in the calibration study were actually “sedentary”. Indeed, while the present study is interesting because it includes commonly used children activity types, there was no measurement of actual METs during the activities. Can the authors ensure that all “sedentary activities” (in particular, seated LEGO) were actually £1.5 METs, as it is currently defined in the literature?

SPECIFIC COMMENTS

Abstract: The term “stationary” should be used in the Objectives and Methods sections of the abstract as it appears in the Results and Conclusions sections.

Line 35: It is written that the MET threshold for sedentary behaviour is “<1.5 METs”, according to reference 3. However, in the cited paper, it is actually indicated “£1.5 METs”. Thus, please use the correct symbol (£).

Lines 61-62: At a first glance, the sentence “none of these studies focussed solely on children’s sedentary behaviours” seems to be an overstatement as several of the cited studies developed raw acceleration thresholds to discriminate “sedentary behaviour” from physical activity in children. However, those studies developed raw acceleration thresholds based on METs data only, not on the activity type performed. I guess that this is the scientific gap that the authors wanted to highlight but as the “sedentary behaviour” concept includes both metabolic and posture components, the statement of the authors evocated above may not be clear enough to explain what it remained to be investigated. Thus, I suggest to provide further explanations/details about this statement and the related literature.

Lines 113-114: It is indicated that start and end times were observed using a Garmin Forerunner235 wristwatch. Please clarify if this watch and all accelerometers were synchronized (to ensure comparable clocks).

Line 100-102: It is written that different ActiGraph accelerometers were used at different body locations. Please explain the rationale for wearing the GT9X on the wrist and the GT3X on the hip and not the inverse, and why both the GT9X and GT3X monitors were not tested both on the wrist and the hip.

Line 135: Did the authors want to write “version 6.13.3” (and not “3.13.3” as it is written)? I ask the question because the version 6.13.3 is the current version of Actilife software (released in 2016).

Line 166: “ActivPAL data were coded two different ways”. Did the authors forget a word between “coded” and “two”?

Line 178: Please indicate why effect sizes were reported as partial eta-squared here and not as as Cohen’s d as reported for other comparisons (cf. line 194).

Data analysis section: I suggest to consider additional statistical approaches to investigate the accuracy of the new cut-points. In particular, authors could also use the following methods to compare ActivPAL results with GA/AG results: correlations, mean percent error (MPE), mean percentage error (MAPE), equivalence testing. These statistical methods have been presented in a paper recently published in the Journal for the Measurement of Physical Behaviour that is headed by leading experts in the field of accelerometer-based measurements (https://journals.humankinetics.com/doi/10.1123/jmpb.2018-0017).

Line 204: Please precise on what is the effect of activity.

Table 3: I have several concerns with this table. First, it is not clear what data have been used to obtain these results. Have the results been obtained from all devices? If this is the case, I don’t see the interest to calculate the means from all body locations and devices. Furthermore, does “Std. Error” means “standard deviation” or “standard error of the mean”? In the latter case, confidence intervals may be enough to indicate uncertainty in the means here.

Line 210: I am not sure that “SE” has been defined before in the manuscript. Please do this.

Section 3.1. Comparison of activities, accelerometer brands and placements: Please indicate in the corresponding tables the differences that were significant to provide a more understandable view of the results. For example, in Table 4, use usual symbols, (*, etc.) to indicate the observed differences. Indicating to the reader the table in which the described results can be found (and doing this before presenting the results) could be a more effective way to present the results (in particular for Table 4).

Table 5: Please define TPR and TNR.

Figure 1: Please modify the figure to ensure that all data points can be visualized. Furthermore, for comparisons between sedentary panels and stationary panels, it could be more accurate to use the same y axis. Please also add the 95% confidence intervals at least for the mean biases and a line corresponding to y=0 to highlight significant over/underestimations. Because one of the aims of the study was to compare GA and AG devices, it could be of interest to construct Bland-Altman plots also for comparing GA and AG estimates (both for sedentary and stationary times).

Line 380: Please cite again the appropriate reference for the concept of Sedentary Sphere.

Author Response

Reviewer 2 – general comments

One of the major limits   is that the reference used to develop the cut-points was the activPAL device,   that did not consistently and correctly classified actual behaviour. As the   authors were present to observe participants during the calibration study, it   should be explained why direct observation was not used as reference to   develop the cut-points. Indeed, while the method used in the study allows   comparisons with previous studies that also used the activPAL, the accuracy   and thus the interest of these cut-points appears limited.

ActivPAL has some inherent limitations, but is viewed as   a valid and reliable method to estimate sedentary time in children and adults (Davies et al. Validity,   practical utility, and reliability of the activpal in preschool children. Med Sci Sport Exer 2012, 44, 761-768; Kozey-Keadle et al.

Validation of wearable   monitors for assessing sedentary behavior. Med Sci Sports Exerc2011;43:1561–7.).

As we were including   free-living data in the analysis, we were unable to complete direct   observation for those periods, therefore observation was not possible.

Observations in the calibration session were not possible   for the transitional movements between stations [the researcher could not   accurately observe all three participants simultaneously within the school   hall]. These periods were included in the calibration protocol as   participants were wearing the activPAL throughout.

From a different study we have observed 96% agreement   between observed behaviours and the activPAL, so we are confident the   majority of behaviours were classified correctly.

We have acknowledged this as a limitation in the   discussion section.

Moreover, only walking   was used as structured active behaviour during the calibration study, while   it could be expected that other active behaviours are used to correspond to   children physical activity patterns.

Yes, walking was the only active behaviour station in the   calibration circuit, but 10 minutes of free-living recess data were included   in the analysis to reflect the range of behaviours children engage in when developing    the thresholds.

Finally, it should be   clarify whether the sedentary behaviours proposed in the calibration study   were actually “sedentary”. Indeed, while the present study is interesting   because it includes commonly used children activity types, there was no   measurement of actual METs during the activities. Can the authors ensure that   all “sedentary activities” (in particular, seated LEGO) were actually £1.5   METs, as it is currently defined in the literature?

Calorimetry was not possible/feasible in this study and we   felt that wearing the equipment would have an impact on ecological validity.

We have added the following line in the limitation   section: “Whilst activities were designed to reflect children’s typical   sedentary behaviours, METs could not be measured and as a result we cannot   conclude that energy expenditure during the protocol were at all times less   than 1.5METs.”

Lines 447-449.

Reviewer   2 – specific comments

Abstract: The term   “stationary” should be used in the Objectives and Methods sections of the   abstract as it appears in the Results and Conclusions sections.

Added “and stationary” Line 17-18

Line 35: It is written   that the MET threshold for sedentary behaviour is “<1.5 METs”, according   to reference 3. However, in the cited paper, it is actually indicated “≤1.5METs”.   Thus, please use the correct symbol ≤

Changed to

Line 35

Lines   61-62: At a first glance, the sentence “none of these studies focussed solely   on children’s sedentary behaviours” seems to be an overstatement as several   of the cited studies developed raw acceleration thresholds to discriminate   “sedentary behaviour” from physical activity in children. However, those   studies developed raw acceleration thresholds based on METs data only, not on   the activity type performed. I guess that this is the scientific gap that the   authors wanted to highlight but as the “sedentary behaviour” concept includes   both metabolic and posture components, the statement of the authors evocated   above may not be clear enough to explain what it remained to be investigated.   Thus, I suggest to provide further explanations/details about this statement   and the related literature.

The cited studies focussed on   the wider physical activity protocol, and did not include more than one or   two sedentary behaviour stations, thus not reflecting the range of sedentary   activities that children participate in. Our study aimed to address this gap   in knowledge by creating a circuit of activities that reflect the range of   sedentary activities children engage in.

Lines 113-114: It is   indicated that start and end times were observed using a Garmin Forerunner235   wristwatch. Please clarify if this watch and all accelerometers were   synchronized (to ensure comparable clocks)

Added: “(synchronized with the same computer time used to   initialise all monitors)”

Line 114

Line 100-102: It is written   that different ActiGraph accelerometers were used at different body   locations. Please explain the rationale for wearing the GT9X on the wrist and   the GT3X on the hip and not the inverse, and why both the GT9X and GT3X   monitors were not tested both on the wrist and the hip.

The redesigned GT9X link   monitor looks like a watch, shows the time and is more appealing to wear at   the wrist, especially for children afraid of standing out amongst their   peers. The GT3X is more cumbersome to wear on the wrist and adding a third   (clunky) monitor to each wrist seemed too burdensome to participants. The two   devices have the same sensors, thus should not record noticeable differences.

We have added: “While the AG   GT9X looks like a watch and shows the time, it is more appealing to wear on   the wrist than the slightly bigger, more cumbersome AG GT3X that is more   suitable for the hip” Lines 130 – 132.

Line 135:   Did the authors want to write “version 6.13.3” (and not “3.13.3” as it is   written)? I ask the question because the version 6.13.3 is the current   version of Actilife software (released in 2016).

Yes, this was written in error.   Thank you for pointing this out.

Changed to 6.13.3 in Line 137.

Line 166: “ActivPAL   data were coded two different ways”. Did the authors forget a word between   “coded” and “two”?

Changed this to “coded in two   different ways”

Line 168

Line 178:   Please indicate why effect sizes were reported as partial eta-squared here   and not as as Cohen’s d as reported for other comparisons (cf. line 194).

Partial eta-squared is most   commonly used for factorial ANOVA designs, mainly due to its availability   through SPSS (Fergusson. An effect size primer: a guide for clinicians and   researchers. Professional Psychology:   Research and Practice. 2009, 40(5),   532-538. DOI: 10.1037/a0015808)

Data   analysis section: I suggest to consider additional statistical approaches to   investigate the accuracy of the new cut-points. In particular, authors could   also use the following methods to compare ActivPAL results with GA/AG   results: correlations, mean percent error (MPE), mean percentage error   (MAPE), equivalence testing. These statistical methods have been presented in   a paper recently published in the Journal for the Measurement of Physical Behaviour   that is headed by leading experts in the field of accelerometer-based   measurements (https://journals.humankinetics.com/doi/10.1123/jmpb.2018-0017).

We have computed as suggested: correlations, MPE, MAPE   and equivalence testing and included the results in the new Table 6 and   Figure 2. We have also highlighted the new text added in Lines 201-203, Lines   275-282, Lines 290-294, Lines 293-296 and 414-417, 458-459.

Line 209:   Please precise on what is the effect of activity.

Apologies, we are not sure   what the reviewer is referring to here. Please clarify and we can address   this comment if required.

Table 3: I have several   concerns with this table. First, it is not clear what data have been used to   obtain these results. Have the results been obtained from all devices? If   this is the case, I don’t see the interest to calculate the means from all   body locations and devices. Furthermore, does “Std. Error” means “standard   deviation” or “standard error of the mean”? In the latter case, confidence   intervals may be enough to indicate uncertainty in the means here.

These are the estimated marginal means from both brands   and both wrists (not the hip data), for each activity.

We have added the following to clarify: “from both wrists   and both brands” to Line 213 and Line 247 (Table heading), as well as “acceleration   (mg)” in the table itself.

Std. Error: removed, just left confidence intervals as   suggested.

Line 210: I am not sure that “SE” has been   defined before in the manuscript. Please do this.

Added: “Standard Error (SE)”

Line 215

Section 3.1. Comparison   of activities, accelerometer brands and placements: Please   indicate in the corresponding tables the differences that were significant to   provide a more understandable view of the results. For example, in Table 4,   use usual symbols, (*, etc.) to indicate the observed differences. Indicating   to the reader the table in which the described results can be found (and   doing this before presenting the results) could be a more effective way to   present the results (in particular for Table 4).

Added symbols to show   significance in Table 4.

Also added the following in   Lines 241-242: “with symbols indicating significant differences between   placements and brands from each activity.”

Table 5: Please define TPR and TNR.

Added “*True Positive Rate  True Negative Rate”

Line 304.

Figure 1: Please modify   the figure to ensure that all data points can be visualized. Furthermore, for   comparisons between sedentary panels and stationary panels, it could be more   accurate to use the same y axis. Please also add the 95% confidence intervals   at least for the mean biases and a line corresponding to y=0 to highlight   significant over/underestimations. Because one of the aims of the study was   to compare GA and AG devices, it could be of interest to construct   Bland-Altman plots also for comparing GA and AG estimates (both for sedentary   and stationary times).

Changed the orientation to landscape. Don’t know why it didn’t   stay as this in the original document.

I tried   using the same y-axis of -400 to +400, but the two top panels end up looking   really squashed and then adding a line corresponding to y=0 makes it hard to   read. I have added the limits of agreement to the new Table 6.

Instead   of cluttering the plots more, we added the limits of agreement in the new   Table 6.

We   added 2 new Bland-Altmans for comparing GA and AG as suggested (Figure 3,   Line 324) and the following sentences in Lines 298-301:  “Figure   3 shows Bland-Altman plots assessing the agreement between sedentary and   stationary time from AG and GA.  Both   had small mean biases (sedentary time = +10.4min; stationary time = +1.4min),   and narrow limits of agreement (sedentary time -19.8min to +40.5min;   stationary time -59.1min to 61.9min).”

Line 380: Please cite again the appropriate reference   for the concept of Sedentary Sphere.

Added. Line 432

Round 2

Reviewer 1 Report

The results presented in figure 1 still require some explanation. The data points in C/D are almost the same, where as the A/B is substantially different. From the text it is stated that A/C is the same device and placement (AG non-dominant wrist) and B/D is the same device and placement (GA non-dominant wrist). If the scatter presented in A/B is so different i would expect the C/D also to be different. But the data point scatter in C/D is almost the same. If this is correct this require an explanation. Furthermore, the number of subjects in A is 21 but in C its only 15 (only 12 in B and 15 in D). There is no mention in the results about loss of subjects, and considering that this is the same device it seem quite strange to loose 6 subjects. Please explain.

Author Response

 Please find our response in the attached Word document.

Reviewer 2 Report

GENERAL COMMENTS

The authors have well considered my comments. In particular, they performed significant modifications of their manuscript by adding complementary results based on additional statistical analyses. These complementary results allow further understanding of the agreement between the different accelerometers outputs. However, some statements in the Results and Discussion sections appear to be no longer suited based on these additional results. Indeed, while initial analyses (i.e., mean biases and mean comparisons) suggested that stationary thresholds are “better” than sedentary thresholds (with lower mean biases and no significant difference compared to activPAL results when using stationary thresholds), the new analyses (in particular MAPE) highlight that stationary thresholds are less precise than sedentary thresholds (with the presence of very large percent individual errors). Actually, this could be already inferred from the initial Bland and Altman analysis showing larger limits of agreement when considering figures for stationary thresholds. This is further highlighted with MAPE results. However, it seems that these larger limits of agreement and MAPE when considering stationary thresholds are caused by two outliers (as shown in Bland and Altman plot). This could be shortly discussed by the authors.

SPECIFIC COMMENTS

Line 203: The names of the authors who are related to the cited reference could be added before “[34]”.

Line 209: In my previous report I wrote “Please precise on what is the effect of activity” and, indeed, this comment was not clear. I apologize for that. This comment suggested indicating that the effect of activity was on the outputs of the accelerometers. Of course this can be inferred from the precedent parts of the article and from the title of the section, but this section begins by “Descriptive data for all participants are shown in Table 2”. This is a bit disturbing, as this information is not related to the title of the section. For the sake of clarity, the authors could add a very short part 3.1. related to the description of the population with very few comments. Then, in the next part, the authors could precise that the presented effects are related to accelerometers outputs, at least at the beginning of the section to ensure the understanding of the reader.

Figures 1 and 3: While the authors have decided to keep the initial y axis for visualization reasons, it still seems to be possible to add a line for y=0 and also 95% confident limits for the mean biases. If possible, I (again) suggest doing this.

Line 269: “Both stationary thresholds performed better in the free-living sample”. I suggest refining this statement as the new analyses can lead to different statements when comparing the error at the group level with the error at the individual level.

Line 405-406: “During the free-living period, the stationary thresholds performed better than the sedentary thresholds ”. Same remark as for line 269 (see above).

Author Response

(The authors gave the same response as above.)

Round 3

Reviewer 1 Report

The new manuscript I have reviewed is with the figure correct. Suggesting that this is not a formatting error during download. However, I still do not feel confident that the data presented in figure 1 is correct. It seems unlikely that a threshold increase of 10 mg with both ActiGraph and GENEActiv can make the scatter go from completely different (sedentary) to almost identical (stationary). The bias goes from a 10 minutes difference (B-A) to only 1.5 minutes difference (D-C). This is two different instruments placed on the same arm and the two differen devices have shown not to generate the exact same output.   

The new manuscript also include another figure 2 and 3. But there is no indication in the manuscript 1 that this should be missing. This is not a download issue but looks like an error during proof reading from the authors side.

Author Response

Answers are in attached file

Reviewer 2 Report

The authors have modified their Bland and Altman plots. However, when I wrote “95% confidence limits” I was not dealing with 95% limits of agreement (it is a good point to show these limits however). I was dealing with the 95% confidence interval for the value of the mean bias (which is different from 95% limits of agreement). The figures would be more informative (and therefore would be better) if 95% confidence intervals for the biases could be added. But this is not yet the case.

Author Response

Please find answers in the attached file

Round 4

Reviewer 1 Report

No further comments to the manuscript.

Author Response

Thanks for your comments